# Gait retraining targeting foot pronation: A systematic review and meta-analysis

**Seyed Hamed Mousavi** [1], **Fateme Khorramroo** [1]*, **Amirali Jafarnezhadgero**[2]

**1** Faculty of Sport Sciences and Health, Department of Sport Injuries and Biomechanics, University of Tehran, Tehran, Iran, **2** Faculty of Educational Science and Psychology, Department of Sport Managements and Biomechanics, University of Mohaghegh Ardabili, Ardabil, Iran

☯ These authors contributed equally to this work.

* negar_moj2004@yahoo.com

**Data Availability Statement:** All relevant data are within the paper.

**Funding:** The author(s) received no specific funding for this work.

## Abstract

Foot pronation is a prevalent condition known to contribute to a range of lower extremity injuries. Numerous interventions have been employed to address this issue, many of which are expensive and necessitate specific facilities. Gait retraining has been suggested as a promising intervention for modifying foot pronation, offering the advantage of being accessible and independent of additional materials or specific time. We aimed to systematically review the literature on the effect of gait retraining on foot pronation. We searched four databases including PubMed, Web of Science, Scopus and Embase from their inception through 20 June 2023. The Downs and Black appraisal scale was applied to assess quality of included studies. Two reviewers screened studies to identify studies reporting the effect of different methods of gait-retraining on foot pronation. Outcomes of interest were rearfoot eversion, foot pronation, and foot arch. Two authors separately extracted data from included studies. Data of interest were study design, intervention, variable, sample size and sex, tools, age, height, weight, body mass index, running experience, and weekly distance of running. Mean differences and 95% confidence intervals (CI) were calculated with random effects model in RevMan version 5.4. Fifteen studies with a total of 295 participants were included. The results of the meta-analysis showed that changing step width does not have a significant effect on peak rearfoot eversion. The results of the meta-analysis showed that changing step width does not have a significant effect on peak rearfoot eversion. Results of single studies indicated that reducing foot progression angle (MD 2.1, 95% CI 0.62, 3.58), lateralizing COP (MD -3.3, 95% CI -4.88, -1.72) can effectively reduce foot pronation. Overall, this study suggests that gait retraining may be a promising intervention for reducing foot pronation; Most of the included studies demonstrated significant improvements in foot pronation following gait retraining. Changing center of pressure, foot progression angle and forefoot strike training appeared to yield more favorable outcomes. However, further research is needed to fully understand its effectiveness and long-term benefits.

**Competing interests:** The authors have declared that no competing interests exist.

## Introduction

Foot pronation is a vital natural movement which involves multiple joint movements [1]. It helps the foot to adopt with the ground [2], contributes in shock absorption [3] and prevents overloading of the lower extremity [2]. Foot pronation contributes to the locking of the tarsal joint, which turns the foot into a rigid lever during late stance and allows it to adjust to uneven terrain. However, it also allows the mid-tarsal joints to unlock, resulting in a more supple, flexible, and pliable forefoot [4]. Rearfoot pronation is accompanied by tibia internal rotation [5] leading to compensatory rotation of femur on transverse plane [6]. Therefore, pronation has an important effect in lower limb kinetic chain and its abnormality may lead to problems in more proximal body parts [7].

Abnormal pronation is determined as a contributing risk factor for running-related injuries (RRIs) [8]. It affects performance, leads to lower limb abnormalities and low back pain [9]. Hence, there has been increasing clinical and scientific interest in interventions targeting atypical foot pronation in order to prevent or manage lower limb injuries [10,11]. These interventions mainly include foot orthoses [10], motion control footwear [12], minimal shoes [13], shoe insoles [14], exercise therapy [14], therapeutic adhesive taping [15], retraining of the intrinsic foot muscles [16], and gait retraining [17]. External supports including foot orthoses [18,19], motion control shoes [19], and therapeutic adhesive taping [20] are the most common interventions studied for reducing over-pronation. Vertical impact peak and peak lateral ground reaction force decreased by ~ 30.5 and ~ 6.2 N during heel contact with the use of single and dual-stiffness shoes in runners with pronated feet [21]. These interventions are widely recommended for correcting atypical foot pronation [18,22]. However, foot orthoses cause dependency [23] and long-term adverse effects (e.g., greater knee abduction [24] and disuse atrophy [25]). Most importantly, their effectiveness is controversial [10]. Only a few studies have previously assessed the effect of a training program on excessive rear-foot eversion [14,26–29], demonstrating sensory-motor training more effective than either foot orthoses [27] or taping for realigning excessive rear-foot eversion [29]. More research into functional training modifying rear-foot eversion is thus warranted.

Gait retraining is increasingly utilized as a novel way of inducing the body or a segment to alter a movement pattern or a segment's motion direction [30]. There is a variety of techniques from easy (e.g., manipulating step rate) to complex (e.g., tibial acceleration decrease) for gait retraining [31]. Recent studies have suggested gait training to change the lower limb biomechanics [27–30,32,33]. The most frequently modified parameters for retraining include step rate, step width and foot strike pattern [33–35].

Several studies have proposed changes to running techniques (i.e., movement) through running retraining using feedback to reduce impact loads [36]. A study found that increasing step cadence by just 5% significantly reduced peak braking force by 5.7% [37] and 11.4% [38] in long-distance runners. Increasing step cadence with a proportional reduction in the stride length at a constant speed has reduced foot inclination angles and impact forces by 5.6% [39] which decreases the number of initial contacts by hindfoot [40]. Besides, altering step width has reduced foot pronation [41,42]. Forefoot strike training has also demonstrated promising results in increasing foot arch [43–45].

Although several studies have assessed the effects of gait retraining on foot pronation, no systematic review synthesizing the evidence on this topic has been published. Therefore, this systematic review and meta-analysis aimed to explore the effect of gait retraining targeting foot pronation. Potential limitations and future research directions are discussed to guide clinical practice and future investigation.

## Method

This systematic review was conducted in accordance with the PERSiST guidelines for systematic reviews [46].

### Search strategy

Relevant studies were identified through four electronic databases: PubMed (1240 studies), Web of Science (2321 studies), Scopus (3430 studies) and Embase (1680 studies). The search was run to extract studies from inception to 20 Jun 2023. Key terms have been used in the search strategy were based on broad terms and related synonyms targeting 2 categories:

#1 pronation OR pronated OR rearfoot OR eversion OR "flat foot" OR "flat feet" OR "pes planus" OR arch OR "foot posture" OR "navicular drop" OR "navicular height"

#2 "foot angle" OR "progression angle" OR "step width" OR retraining OR speed OR "toe-in" OR "in-toeing" OR "toeing-in" OR "toe-out" OR out-toeing OR toeing-out OR "strike pattern" OR "backward walking" "COP" OR "center of pressure" OR "foot adduction" OR "foot abduction"

#1 AND #2

Reference lists from previous related systematic reviews on gait retraining targeting foot pronation were hand searched to ensure the identification of all relevant studies.

### Eligibility criteria

All searches were conducted separately based on established guidelines for inclusion criteria and extraction forms.

The inclusion criteria were: Written-English studies comparing the effect of gait retraining before and after interventions on foot pronation, in studies that included participants with either supinated or pronated foot or participants without any abnormality in the foot arch.

The exclusion criteria were: non-English studies, studies with an intervention other than gait retraining or assessing effects other than foot pronation or investigated on individuals with specific abnormalities such as knee valgus.

### Study selection

Two authors (FKH and SHM) independently screened the title, abstract and full-text of studies, in line with the inclusion criteria. In any case of disagreements, a consensus was reached by discussion of 2 reviewers or third reviewer (AJG) if needed.

### Quality assessment

Two authors (FKH and SHM) independently assessed the methodological quality of the included studies using the modified Downs and Black checklist [47]. The complete form was used to assess RCTs and 15 questions were used to assess non-RCTs. Any disparities in scoring were rechecked and if necessary, a consensus was reached using the third reviewer (AJG).

### Data collection

One author (FKH) extracted all relevant data from the included studies. In order to minimize potential bias or inaccuracies in the data collection process, all data was cross-checked by (SHM). In this review, ankle kinematic data related to foot pronation were extracted. Information from study design, intervention, outcomes, sample size, sex, tools, age, height, weight,

**Table 1. Definitions of modified level of evidence.**

| Level of evidence | Description |
|---|---|
| Strong evidence | Pooled results from three or more studies, including a minimum of two high-quality studies which are statistically homogenous (p>0.05)- may be associated with a statistically significant or non-significant pooled results. |
| Moderate evidence | Statistically significant pooled results from multiple studies, including at least one high-quality study, which are statistically heterogeneous (p<0.05); or from multiple low- or moderate-quality studies which are statistically homogenous (p>0.05); or statistically insignificant pooled results from multiple studies, including at least one high-quality study, which are statistically homogenous (p>0.05). |
| Limited evidence | Results from multiple low- or moderate-quality studies which are statistically heterogeneous (p<0.05); or from one high-quality study. |
| Very limited evidence | Results from one low- or moderate-quality study. |
| Conflicting evidence | Pooled results that are insignificant and from multiple studies, regardless of quality, which are statistically heterogeneous (p<0.05, i.e., inconsistent). |

body mass index (BMI), running experience, and weekly distance were extracted from the included studies.

## Synthesis of results

Mean differences and 95% confidence intervals (CI) were calculated with random effects model in RevMan version 5.4. A meta-analysis was conducted when a minimum of 2 studies examined the same outcome measure using similar methodologies. The level of statistical heterogeneity for pooled data was quantified by $I^2$ statistics and related P-values (P<0.05). Results were achieved by means of levels of evidence as defined by van Tulder et al., [48] modified by Mousavi et al., [49] Table 1.

## Results

The main literature search yielded a total of 4562 from which 1702 items remained after duplicate removal. A total of 1688 studies were excluded due to not meeting the inclusion criteria and 15 were included after screening of titles and abstracts for further eligibility check [17,37,41–45,50–57]. (Fig 1) shows the flow diagram, summarizing the selection process and the number of studies excluded at each stage.

### Study characteristics

Table 2 summarizes the characteristics of the included studies. There were 12 cross-sectional studies [17,37,41,42,44,45,50–54,57] and 3 RCTs [43,52,58] assessing the effects of gait retraining on foot pronation.

### Quality assessment

Fourteen studies were assessed by Downs and black scale [17,21,37,52,55–57,59]. Any disparities in scoring were rechecked by 2 authors (FKH and SHM). Table 3. shows the results of quality assessment. The average score of eligible studies was 13.09 for cross-sectional studies [17,37,41,42,44,45,50,51,54–57] and 21 for the RCTs [55,60,61]. All studies outcomes were reported from more than 85% of the subjects initially allocated to treatment or control group.

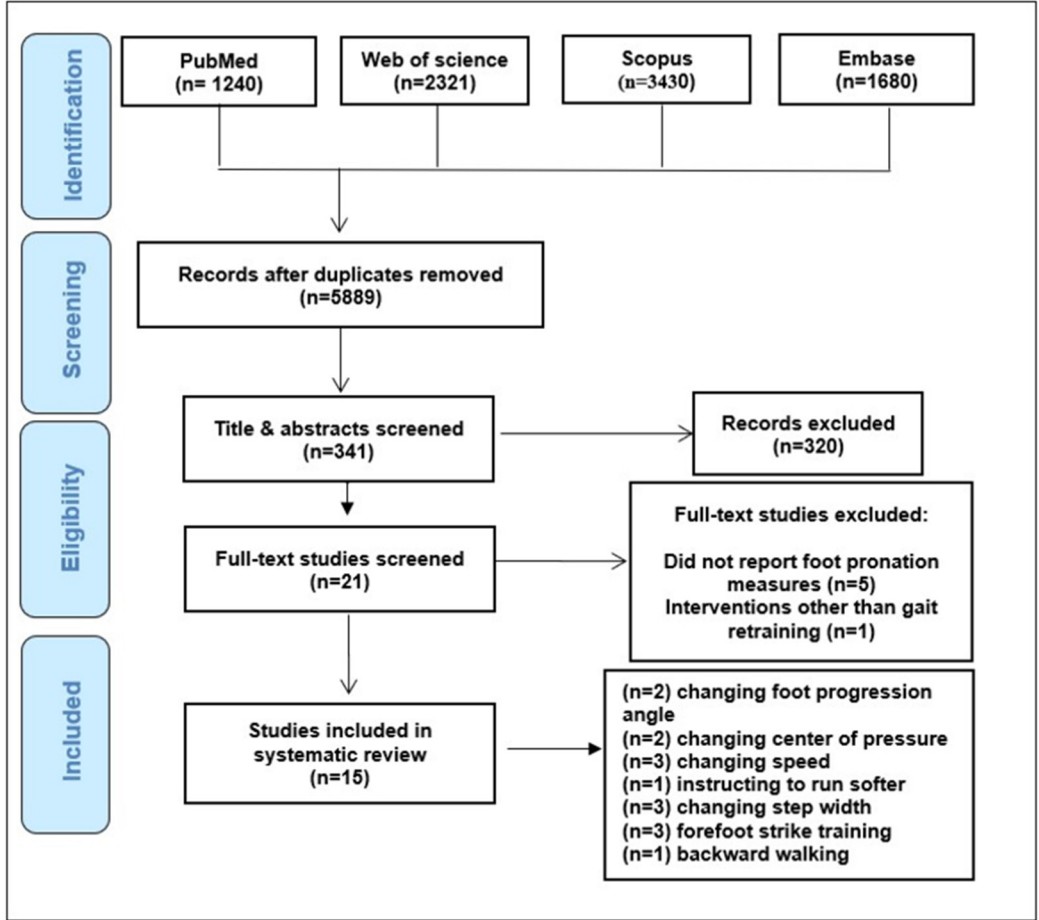

**Fig 1. Flow chart of study selection process.**

## Changing step width

Three studies assessed changing step width (normal, wide and narrow) [41,42,50]. The results of meta-analysis are shown in (Fig 2). The results of the meta-analysis were not significant for peak rearfoot eversion angle while decreasing or increasing step width.

Increasing step width from narrow to wide decreased rearfoot eversion angles. [41].

Rearfoot kinematics were significantly different from normal running and cross-over (narrow) running. Peak rearfoot eversion angle was greater in narrow walking by -1.40° [-3.96, 1.16] and peak rearfoot eversion excursion by -2.80° [-6.97, 1.37] and time to peak rearfoot eversion happened 6–7% earlier in stance phase (P<0.05) but not in wide running [42].

Decreasing step width, increased maximum pronation from normal to wide and narrow by -3.70° [-6.70, -0.70] and 2.40° [-1.14, 5.94], pronation from normal to wide and narrow by -3.60 [-8.56, 1.36] and 3.40° [-0.48, 7.28] and maximal pronation velocity by -95.80° [-235.02, 43.42] and 110.20° [-48.66, 269.06] [50]. (Fig 3).

## Changing foot progression angle

Two studies assessed the effect of changing foot progression angle on rearfoot kinematics [62,63]. In the study by Mousavi et al. [62] subjects performed toe-in/toe-out running using

**Table 2. Characteristics of the included studies.**

| Study | Study design | Intervention | Variable | Sample size and sex | Tools | Age (y) | Height (cm) | Weight (kg) | Body mass index (kg/m²) | Running experience (y) | Weekly distance (Km) |
|---|---|---|---|---|---|---|---|---|---|---|---|
| Pohl et al., 2007 [54] | Cross-sectional | Changing speed | Rearfoot eversion excursion | 12 subjects 6(m), 6(f) | seven ProReflex cameras (Qualisys Medical AB, Sweden) | 22.6 (4.0) | 171.9 (8.6) | 63.0 (10.8) | | | engaging in at least 2 h per week of exercise involving running |
| Mousavi et al., 2021 [17] | Cross-sectional | Real-time visual feedback (Toe-in & toe-out | Peak rearfoot eversion, Time to peak rearfoot eversion (% stance), Rearfoot eversion at touchdown, Rearfoot eversion excursion, Peak pronation, Time to peak pronation (% stance), Supination/pronation at touchdown, Pronation excursion Peak MLAA, Time to peak MLAA (% stance), MLAA at touchdown, MLAA excursion | 17 runners (F) | Instrumented split-belt treadmill with two integrated force plates of the Gait Real-time Analysis Interactive Lab (GRAIL) system (Motekforce Link, The Netherlands), 10-camera integrated motion capture system (Vicon Bonita 10; Vicon Motion Systems, Oxford, UK), D-Flow (Version 3.28; Motekforce Link, The Netherlands) | 21–40 | 164–182 | 50–72 | (18.59–21.73) | 2–17 | 10–65 |
| Charlton et al., 2018 [55] | Experimental single session repeated measure | Changing foot progression angle | Frontal plane rearfoot angle at IC, peak frontal angle in stance, frontal rearfoot excursion, ankle inversion | 6(M), 9(F) with medial compartment knee Osteoarthritis | 14-camera high-speed motion analysis system (Motion Analysis Corp, Santa Rosa, CA), force platform (Advanced Mechanical Technology Inc, Watertown, MA) embedded in the center of a 10-m wooden walkway. | 67.9 ± 9.4 | 167 ± 11 | 75.6 ± 15.0 | (24.90–28.59) | 0 | 0 |

*(Continued)*

**Table 2.** (Continued)

| Study | Study design | Intervention | Variable | Sample size and sex | Tools | Age (y) | Height (cm) | Weight (kg) | Body mass index (kg/m²) | Running experience (y) | Weekly distance (Km) |
|---|---|---|---|---|---|---|---|---|---|---|---|
| Farina and Hahn, 2022 [37] | Cross-sectional | Increasing step rate | Peak eversion | 11(M), 9(F) runners | 8-camera motion capture system (Motion Analysis Corp., Rohnert Park, CA, USA), instrumented treadmill (Bertec, Columbus, OH, USA), Standardized, neutral running shoes (Brooks Launch) | 24.9 ± 8.66 | 173.69 ± 9.83 | 64.69 ± 11.27 | (24.90–28.59) | | 34.50 ± 17.08 |
| Dunn et al., 2018 [57] | Cross-sectional | Changing speed | Peak foot eversion, foot eversion excursion | Int: 6(M), 4(F) Cont: 4(M), 6(F) Uninjured | Eight camera, digital motion capture system sampling at 200 Hz (Motion Analysis Corporation, Santa Rosa, CA, USA), force platform (9281CA, Kistler Instrumente, AG, Switzerland), laboratory-based treadmill (Saturn, H-P-Cosmos Sports & Medical, GmbH, Germany) | Int (29.5 ± 3.9) Cont (29.3 ± 3.4) | Int (170 ± 16) Cont (174 ± 9) | Int (69.4 ± 9.6) Cont (69.2 ± 10.9) | Int (22.83–25.21) Cont (21.41–23.91) | | |
| Silva Neto et al, 2021 [52] | RCT | Running softer | Medial longitudinal arch, plantar arch index | 24 recreational runners. int: 58.3% (F); 41.6% (M) cont: 66.7% (F); 33.3% (M) Recreational runners | Pressure platform (Loran ®, Artigianale BO, Italy, 700L), | Int (44.0 ±8.9) Cont (44.2 ±8.1) | Int (170±10) Cont (160 ± 10) | Int (69.1 ±10.3), Cont (65.1 ±7.5) | Int (20.34–27.47), cont (22.5–28.35) | Int (18.5 ±1.2), Cont (19.0±1.0) (mo) | ≥20 |

*(Continued)*

**Table 2.** (Continued)

| Study | Study design | Intervention | Variable | Sample size and sex | Tools | Age (y) | Height (cm) | Weight (kg) | Body mass index (kg/m²) | Running experience (y) | Weekly distance (Km) |
|---|---|---|---|---|---|---|---|---|---|---|---|
| Mousavi et al., 2021 [56] | Cross-sectional | Changing mediolateral COP | FPA and ML COP in midstance, Peak rearfoot eversion, Time to peak rearfoot eversion (% stance), Rearfoot eversion at TD, Rearfoot eversion excursion, Peak subtalar pronation, Time to peak pronation (% stance), Subtalar pronation at TD, Subtalar pronation excursion, Peak MLAA, Time to peak MLAA (% stance), MLAA at TD, MLAA excursion | 17 runners(F) | Instrumented split-belt treadmill with two integrated 3D force plates of the Gait Real-time Analysis Interactive Lab (GRAIL) system (Motekforce Link, The Netherlands) synchronized with a 10-camera integrated motion capture system (Vicon Bonita 10; Vicon Motion Systems, Oxford, UK) | 21– 40 | 164– 182 | 50 – 72 | (18.59– 21.73) | 2– 17 | 10– 65 |
| Brindle et al., 2014 [41] | Cross-sectional | Changing step width | Rearfoot eversion angle | 26 (M), 25 (F) Healthy | Nine-camera motion capture system (Vicon, Oxford, UK), force plate (Advanced Mechanical Technology, Inc., Watertown, MA, USA) | 26±3 (M), 25±5 (F) | 1.77±0.05 (M), 1.65 ±0.06 (F) | 73.94±4.27 (F), 59.03 ±6.60 (M) | | | 35±17 (M), 25 ±10 (F) |
| Pohl et al., 2006 [42] | Cross-sectional | Changing step width | Rearfoot peak eversion and excursion, time to peak rearfoot everaion | 6 (M), 6 (F) | Seven ProReflex cameras (Qualisys Medical AB, Gothenburg, Sweden) | 29.9±4.9 | 1.71±0.95 | 61.2±15.1 | | | |

(*Continued*)

**Table 2.** (Continued)

| Study | Study design | Intervention | Variable | Sample size and sex | Tools | Age (y) | Height (cm) | Weight (kg) | Body mass index (kg/m²) | Running experience (y) | Weekly distance (Km) |
|---|---|---|---|---|---|---|---|---|---|---|---|
| Shen et al., 2012 [43] | RCT | Forefoot strike training | Arch angle, max arch height, arch height at touchdown, arch height, arch stiffness | 26 (M) recreational, habitual rearfoot strike runners | 12-camera motion analysis system (100Hz, Vicon Motion Analysis Inc., Oxford, United Kingdom), 90 × 60 × 10 cm force platforms (1000 Hz, 9287 B, Kistler Corporation, Winterthur, Switzerland) | Int (25.2 ±4.8), cont (23.8±1.7) | Int (175.5 ±8.2), cont (176.6±4.9) | Int (72.8 ±14.9), cont (72.0±7.1) | | | Weekly frequency; 2.8 (0.4), Int 29.4 (4.3), Cont 28.8 (3) |
| Williams and Ziff, 1991 [50] | Cross-sectional | Changing step length, width, shoulder rotation degree | Pronation | 8 (M) runners | High-speed cine film collected using a LOCAM camera operating at 200 fps and an additional camera running at 50 fps | | | | | | |
| Laughton et al., 2003 [45] | Cross-sectional | Forefoot strike training | Rearfoot excursion | 15 injury free recreational rearfoot strike runners | Nike Air Pegasus shoes (Nike, Beaverton, OR), uniaxial accelerometer (model 353B17), a sensor signal conditioner (model 480E09 ICD), 10 (PCB Pieziotronics, Depew, NY) 64-channel 12-bit A/ D board (Vicon Motion Systems, Lake Forest, CA) | 22.46 ± 4 yrs | 169.75 ± 6.07 cm | 66.41 ± 8.58 kg | | | |
| Williams et al., 2000 [44] | Cross-sectional | Forefoot strike training | Inversion at foot strike, eversion excursion, eversion velocity, ankle inversion moment | 18 recreational runners: 9 rearfoot and 9 forefoot strikers, 6 (M) and 3 (F) in each group | (Bertec, OH) forceplate, 5 vicon cameras (Oxford Metrics, UK) | 18–45 | | | | | |

(*Continued*)

**Table 2.** (Continued)

| Study | Study design | Intervention | Variable | Sample size and sex | Tools | Age (y) | Height (cm) | Weight (kg) | Body mass index (kg/m²) | Running experience (y) | Weekly distance (Km) |
|---|---|---|---|---|---|---|---|---|---|---|---|
| Saleh et al., 2022 [53] | RCT | Backward walking training | Foot posture index | 44: 37 (F), 7 (M) with mobile flat foot | HUMAC Balance System; a computerized dynamic posturography device (Stoughton, MA, USA, 2013) in which the force platform sensors measure the forces produced by the technology of the Wii balance boards. | 19–35 | Int (164.22 ±8.08), cont (164.77±9.7) | Int (61.04 ±6.87), cont (62.4±9.88) | Int (22.58 ±1.41), Cont (22.58 ±1.73) | | |
| Browne 2016 [51] | Cross-sectional | Changing mediolateral COP | Rearfoot eversion | 8 healthy, 5 (M), 3 (F) | Monitor, instrumented treadmill, motion capture (vicon nexus) | 26.9 (2.7) years | | 70.49±13.57 kg | | | |

Notes: FPA, Foot progression angle; ML, Mediolateral; COP, Center of pressure; TD, Touch down; F, female; M, male; MLAA, Medial Longitudinal Arch Angle; IC, initial contact; yrs, years; mo, month; int, intervention; cont, control.

**Table 3. Results of quality assessment.**

| Questions | 1 Aim clearly described? | 2 Main outcomes described in introduction or method? | 3 patient's characteristics clearly described? | 4 Interventions clearly described? | 5 Principal confounders clearly described | 6 Main findings clearly described? | 7 Estimates of random variability provided for main outcomes? | 8 All adverse events reported? * | 9 Characteristics of patients lost to follow up described? | 10 p-value report for main outcome? | 11 Subjects asked to participate representative of source population? | 12 Subjects prepared to participate representative of source population? | 13 Location and delivery of treatment was representative of source population? * | 14 Study participants blinded to treatment? |
|---|---|---|---|---|---|---|---|---|---|---|---|---|---|---|
| Pohl et al., 2007 [52] | 1 | 1 | 1 | | 1 | 1 | 1 | | | 1 | 0 | 0 | | |
| Mousavi et al., 2021 [18] | 1 | 1 | 1 | | 2 | 1 | 1 | | | 1 | 1 | 0 | | |
| Charlton et al., 2018 [53] | 1 | 1 | 1 | | 1 | 1 | 1 | | | 1 | 1 | 0 | | |
| Farina and Hahn, 2022 [38] | 1 | 0 | 1 | | 2 | 1 | 1 | | | 1 | 1 | 0 | | |
| Silva Neto et al., 2021 [50] | 1 | 1 | 1 | 1 | 2 | 1 | 1 | 0 | 1 | 1 | 0 | 0 | 0 | 0 |
| Dunn et al., 2018 [58] | 1 | 1 | 1 | | 1 | 1 | 1 | | | 1 | 1 | 0 | | |
| Mousavi et al., 2021 [54] | 1 | 1 | 1 | | 2 | 1 | 1 | | | 1 | 1 | 0 | | |
| Brindle et al., 2014 [42] | 1 | 1 | 1 | | 1 | 1 | 1 | | | 1 | 0 | 0 | | |
| Pohl et al., 2006 [43] | 1 | 1 | 1 | | 1 | 1 | 1 | | | 1 | 0 | 0 | | |
| Shen et al., 2012 [44] | 1 | 1 | 1 | 1 | 2 | 1 | 1 | 0 | 1 | 1 | 0 | 0 | 1 | 0 |
| Williams and Ziff 1991 [56] | 1 | 1 | 1 | | 1 | 1 | 1 | | | 1 | 0 | 0 | | |
| Laughton et al., 2003 [46] | 1 | 1 | 1 | | 2 | 1 | 1 | | | 0 | 1 | 0 | | |
| Williams et al., 2000 [45] | 1 | 1 | 1 | | 1 | 1 | 1 | | | 0 | 0 | 0 | | |
| Browne 2016 [57] | 1 | 1 | 1 | | 1 | 1 | 0 | | | 0 | 0 | 0 | | |
| Saleh et al., 2022 [51] | 1 | 1 | 1 | 1 | 2 | 1 | 1 | 0 | 0 | 0 | 1 | 1 | 1 | 0 |
| Percentage agreement reliability | 100% | 100% | 92% | | 96% | 100% | 96% | | | 100% | 96% | 92% | | |

| Questions | 15 Blinded outcome assessment | 16 Any data dredging clearly described? * | 17 Analysis adjusts for differing follow-up length? | 18 Appropriate statistical test performed? | 19 Compliance with interventions was reliable? * | 20 Outcome measures were reliable and valid? | 21 All participants recruited from the source population? * | 22 All participants recruited over the same period of time? | 23 Participants randomized treatment? | 24 Allocation of treatment concealed from investigators and participants? | 25 Adequate adjustment for confounding? | 26 Losses to follow up taken into account? | 27 Sufficient power to detect treatment effect at significance level of 0.05? | Total |
|---|---|---|---|---|---|---|---|---|---|---|---|---|---|---|
| Pohl et al., 2007 [52] | 1 | 1 | 1 | 1 | | 1 | 1 | 0 | | 0 | 0 | | | 11 |

(Continued)

**Table 3.** (Continued)

| Study | | | | | | | | | | | | | Score |
|---|---|---|---|---|---|---|---|---|---|---|---|---|---|
| Mousavi et al., 2021 [18] | 1 | | 1 | | 1 | | 0 | 1 | 1 | 0 | | 1 | 14 |
| Charlton et al., 2018 [53] | 1 | | 1 | | 1 | | 0 | 1 | 1 | 0 | | 0 | 12 |
| Farina and Hahn, 2022 [38] | 1 | | 1 | | 1 | | 0 | 0 | 1 | 0 | | 1 | 12 |
| Silva Neto et al., 2021 [50] | 1 | 1 | 1 | 1 | 1 | 1 | 0 | 1 | 1 | 1 | 0 | 1 | 20 |
| Dunn et al., 2018 [58] | 1 | | 1 | | 1 | | 0 | 1 | 1 | 0 | | 1 | 13 |
| Mousavi et al., 2021 [54] | 1 | | 1 | | 1 | | 1 | 1 | 1 | 1 | | 1 | 15 |
| Brindle et al., 2014 [42] | 1 | | 1 | | 1 | | 0 | 1 | 1 | 0 | | 1 | 12 |
| Pohl et al., 2006 [43] | 1 | | 1 | | 1 | | 0 | 0 | 1 | 0 | | 1 | 11 |
| Shen et al., 2012 [44] | 1 | 1 | 1 | 1 | 1 | 1 | 0 | 1 | 1 | 1 | 0 | 1 | 22 |
| Williams and Ziff 1991 [56] | 1 | | 1 | | 1 | | 0 | 1 | 1 | 0 | | 0 | 11 |
| Laughton et al., 2003 [46] | 1 | | 1 | | 1 | | 0 | 1 | 1 | 0 | | 1 | 13 |
| Williams et al., 2000 [45] | 1 | | 1 | | 1 | | 0 | 0 | 1 | 0 | | 0 | 9 |
| Browne 2016 [57] | 1 | | 1 | | 1 | | 0 | 1 | 1 | 0 | | 1 | 10 |
| Saleh et al., 2022 [51] | 1 | 1 | 1 | 0 | 1 | 1 | 1 | 1 | 1 | 1 | 0 | 1 | 21 |
| Percentage agreement reliability | 95% | | 100% | | 100% | | 100% | 92% | 96% | 100% | | 100% | |

Key: 1 = Yes; 0 = No. *2 = Yes; 1 = Partially; 0 = No; * = the question discussed with the third reviewer.

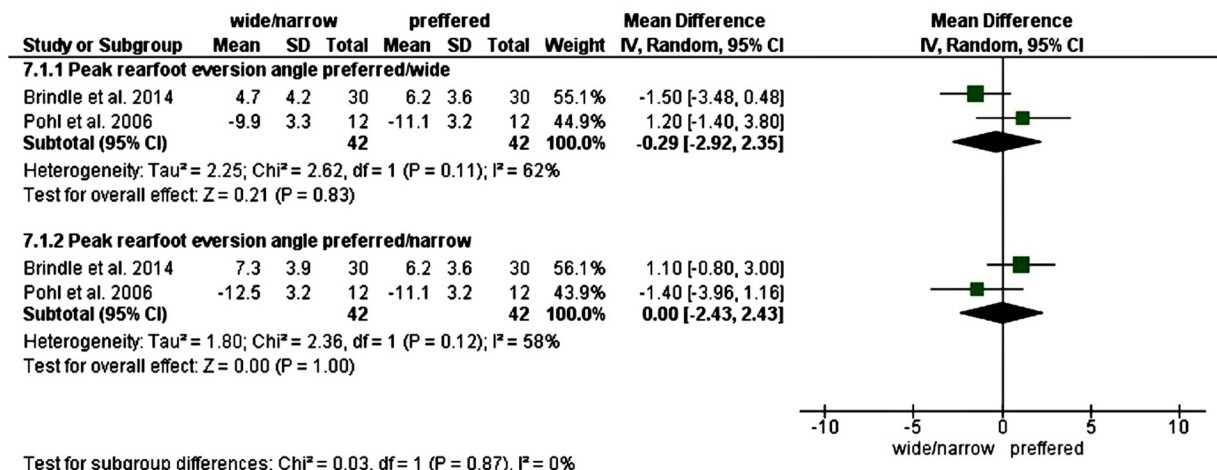

**Fig 2. Results of meta-analysis for changing step width on peak rearfoot eversion.**

real-time visual feedback which was set ±5˚ from habitual foot progression angle. In the study by Charlton et al. [63] subjects walked in 4 conditions guided by real-time biofeedback: (1) toe-in (+10˚), (2) zero rotation (0*), (3) toe-out (-10˚), and (4) toe-out (-20˚).

Toe-in running decreased peak rearfoot eversion, peak pronation, and peak medial longitudinal arch angle (MLAA) in healthy runners [17]. In patients with osteoarthritis, toe-in walking reduced peak rearfoot eversion and increased rearfoot inversion at initial contact, rearfoot excursion in frontal plane, peak external ankle inversion moments. In contrast, walking with 20 degrees of toe-out significantly reduced rearfoot inversion angles [55] (Figs 4 and 5).

## Changing speed

Three studies investigated the effect of changing speed on rearfoot kinematics [37,42,57]. In the study by Dunn et al. [57] all participants ran at relative (REL: 1.5 km/h −1 below respiratory compensation point) and absolute (ABS: 4.5 m·s−1) speeds. In the study by Farina and Hahn [37] participants ran at their preferred pace and step rate, then +5% and +10% of their preferred step rate while being cued by a metronome. In the study by Pohl et al. [42] subjects walked/ran barefoot over-ground at one walking (50% maximum walking speed in which they could not refrain from running) and three running speeds (slow, 120 and 140% of maximum walking speed).

Running retraining in the study by Pohl et al. [54] was significant only between walking and fast running.

Five percent and 10% increase in step rate reduced peak rearfoot eversion. In the +5% condition between 30.8% and 42.1% of stance by 0.57° [-2.71, 3.85] and in the +10% condition between 20.4% and 44.0% of stance by 0.79° [-2.52, 4.10] as compared with the preferred condition [37]. (Fig 6).

Pose method had a non-significant decrease in peak rearfoot eversion angle by 1.61° [-2.12, 5.34] for relative speed and by 1.90° [-2.23, 6.03] for absolute speed [57]. (Fig 7).

## Running softer

The study by Silva Neto et al. [52] measured rearfoot kinematics after an 8-week gait retraining program in which the participants were asked to "run softer" so that the amplitude of the vertical impact peak (maximum force) would be reduced.

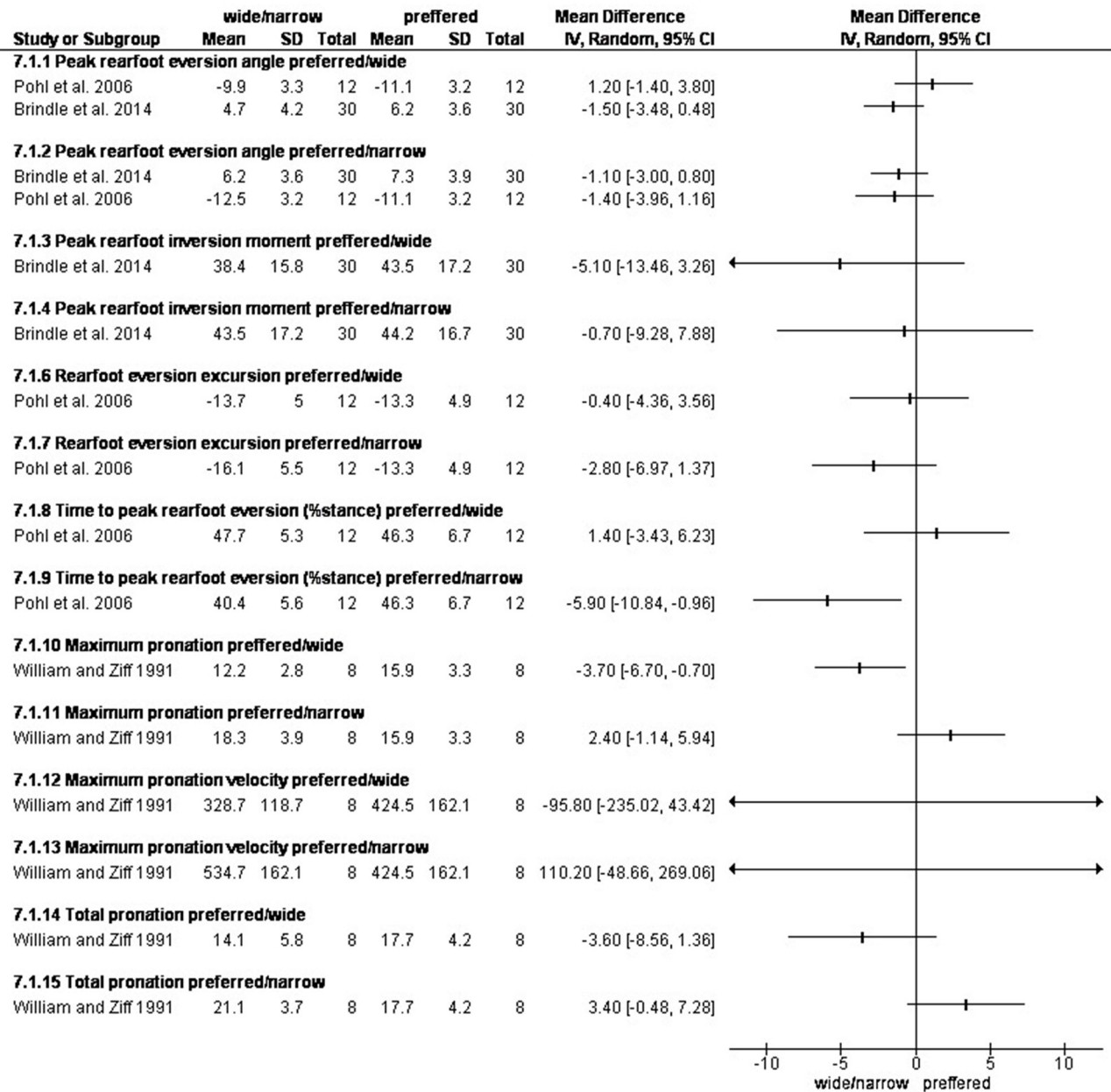

**Fig 3. Results of changing step width on rearfoot pronation/eversion.**

A 2-week gait retraining program using visual biofeedback (instructed to run softer), increased left and right foot MLAA by 0.40° [-0.09, 0.89] and 0.50° [0.01, 0.99] and left and right plantar arch index (AI) (dynamic arch) by -0.02 [-0.48, 0.44] and -0.01 [-0.45, 0.43] [52]. The dynamic arch index was increased in runners who had a reduced AI before the intervention, indicating a foot cavus with an adjustment in plantar support resulting in an increase in the AI after the intervention [52] (Fig 8).

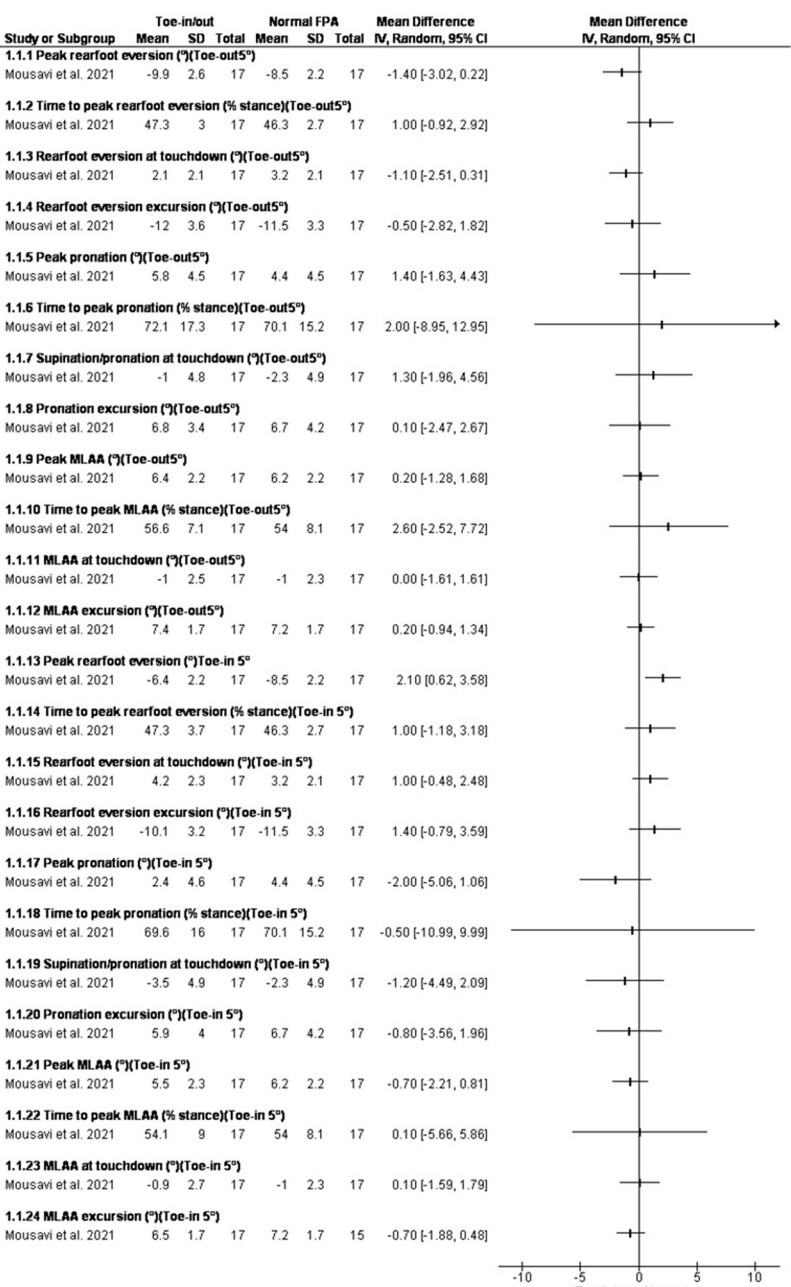

**Fig 4. Results of Changing foot progression angle on rearfoot eversion and MLAA.**

## Changing center of pressure

Two studies investigate the effect of changing [51,60]. In the study by Mousavi et al. [60] subjects ran with normal, medial and lateral COP, while foot progression angle was controlled using visual feedback. In the study by Brown et al. [51] subjects walked on an instrumented treadmill while provided bilateral visual biofeedback targets for toe-off on a visual display alongside real-time COP trajectories. Toe-off targets included a neutral location along with medial, lateral, anterior and posterior shifts.

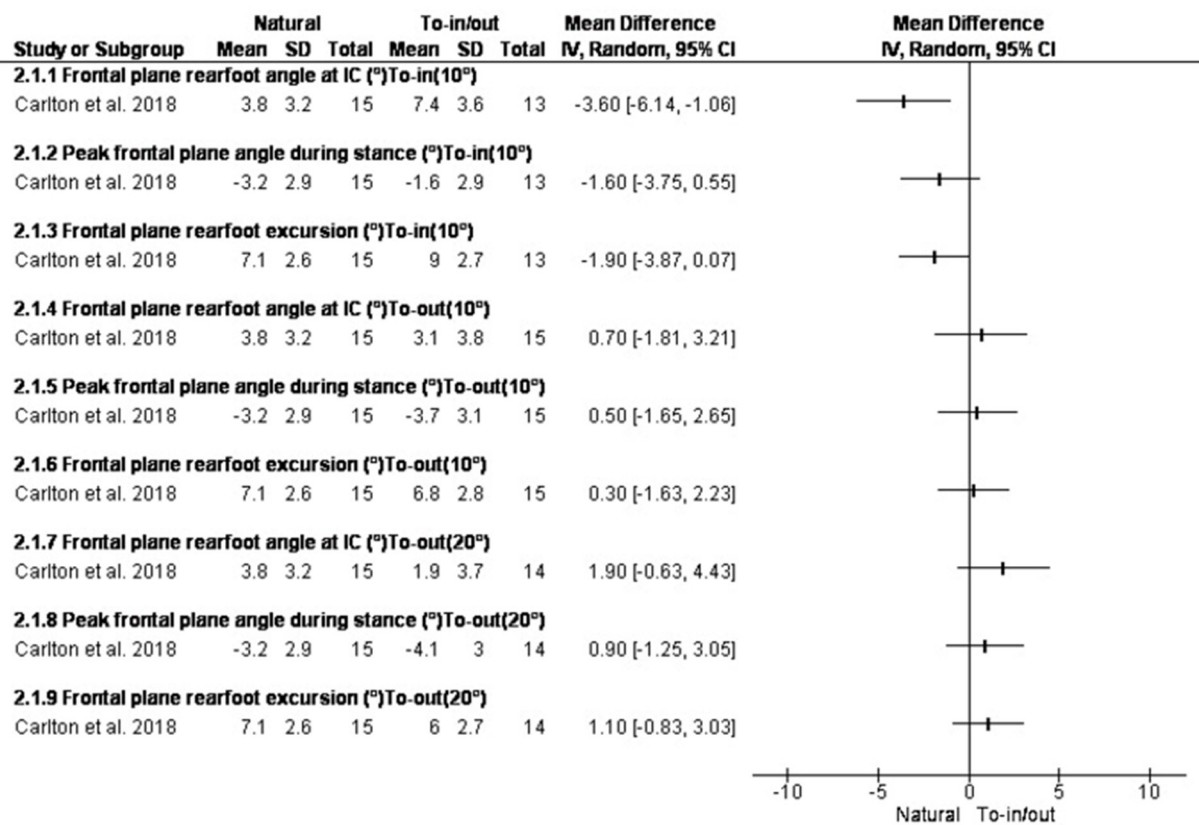

**Fig 5. Results of changing foot progression angle by 10 degrees on rearfoot eversion.**

Running with more lateral COP decreased peak rearfoot eversion by -3.30º [-4.88, -1.72], peak subtalar pronation by 5.00º [2.85, 7.15], and peak MLAA by 2.30º [0.79, 3.81], respectively, compared to normal running (toe-out is negative) [56]. Running with more lateral COP increased peak rearfoot eversion by 2.70º [1.15, 4.25], peak subtalar pronation by -4.60º [-8.13, -1.07], peak MLAA by -1.80º [-3.28, -0.32] respectively, compared to normal running (toe-out is negative) [56]. Spatial modifications to the progression of the COP resulted in a laterally or medially shifted COP, which led to changes in peak inversion ankle angle and moment [51] (Fig 9).

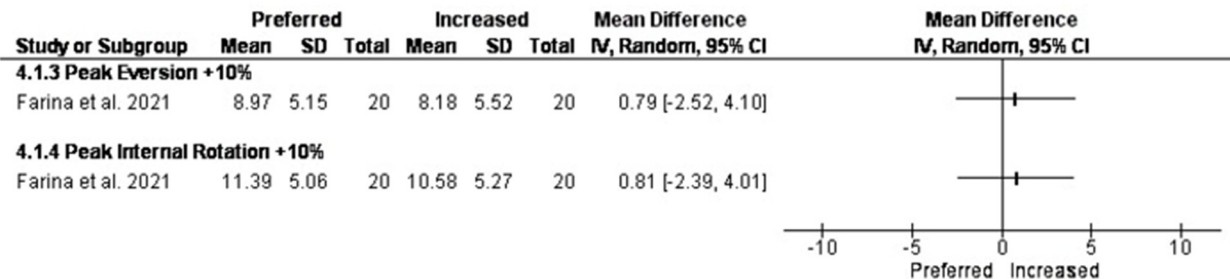

**Fig 6. Results of changing step rate on rearfoot eversion.**

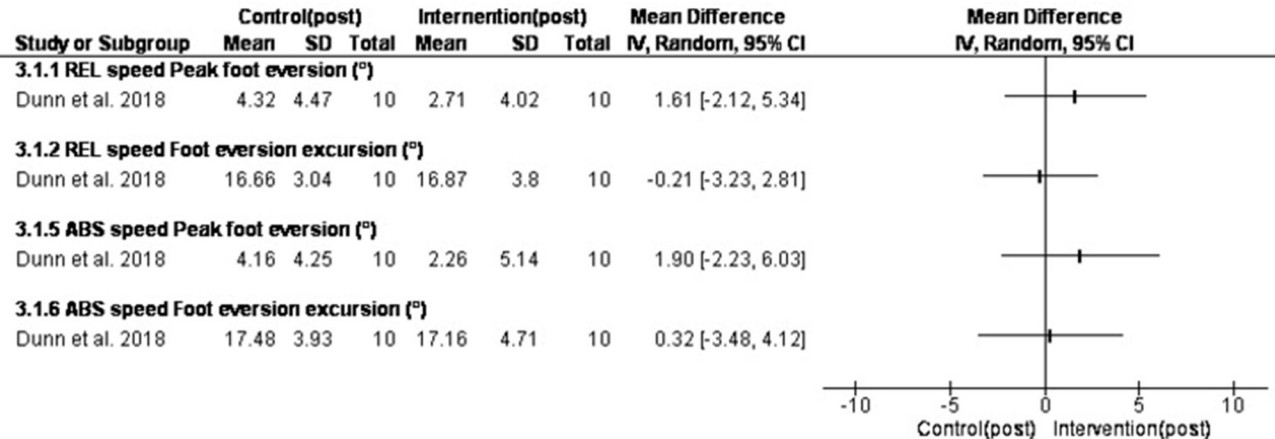

**Fig 7. Results of pose training (changing speed) on rearfoot eversion.**

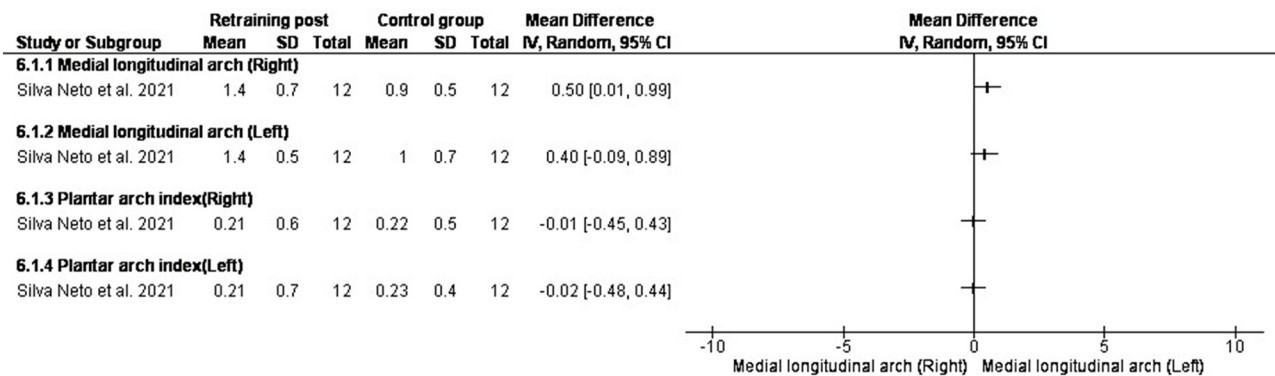

**Fig 8. Results of running softer on MLAA.**

## Changing foot strike

Three studies investigated the effect of forefoot strike training. Shen et al. [43] investigated the effect of a 12-week forefoot strike training combined with foot core exercise and Laughton et al. [45] assessed forefoot and rearfoot strike while William III [44] compared forefoot strikers with those who were instructed to run with a forefoot strike pattern.

Forefoot strike (FFS) significantly increased normalized navicular height by 5.1% and arch height by 32.1% at touch down [43]. FFS significantly increased eversion excursion by 2.72° [-0.15, 5.59], due to greater amounts of inversion and plantarflexion at foot strike with the FFS pattern [45].

'Rearfoot kinematics were not significantly different between FFS and instructed forefoot strike runners (CFFS). The CFES were inverted at foot strike. They also demonstrated similar rearfoot eversion excursions. Eversion velocities of the CFFS were similar to the FFS [44] (Fig 10).

## Backward walking

Saleh et al. [58] investigated the effect of backward walking in flat-footed individuals before and after 18 sessions.

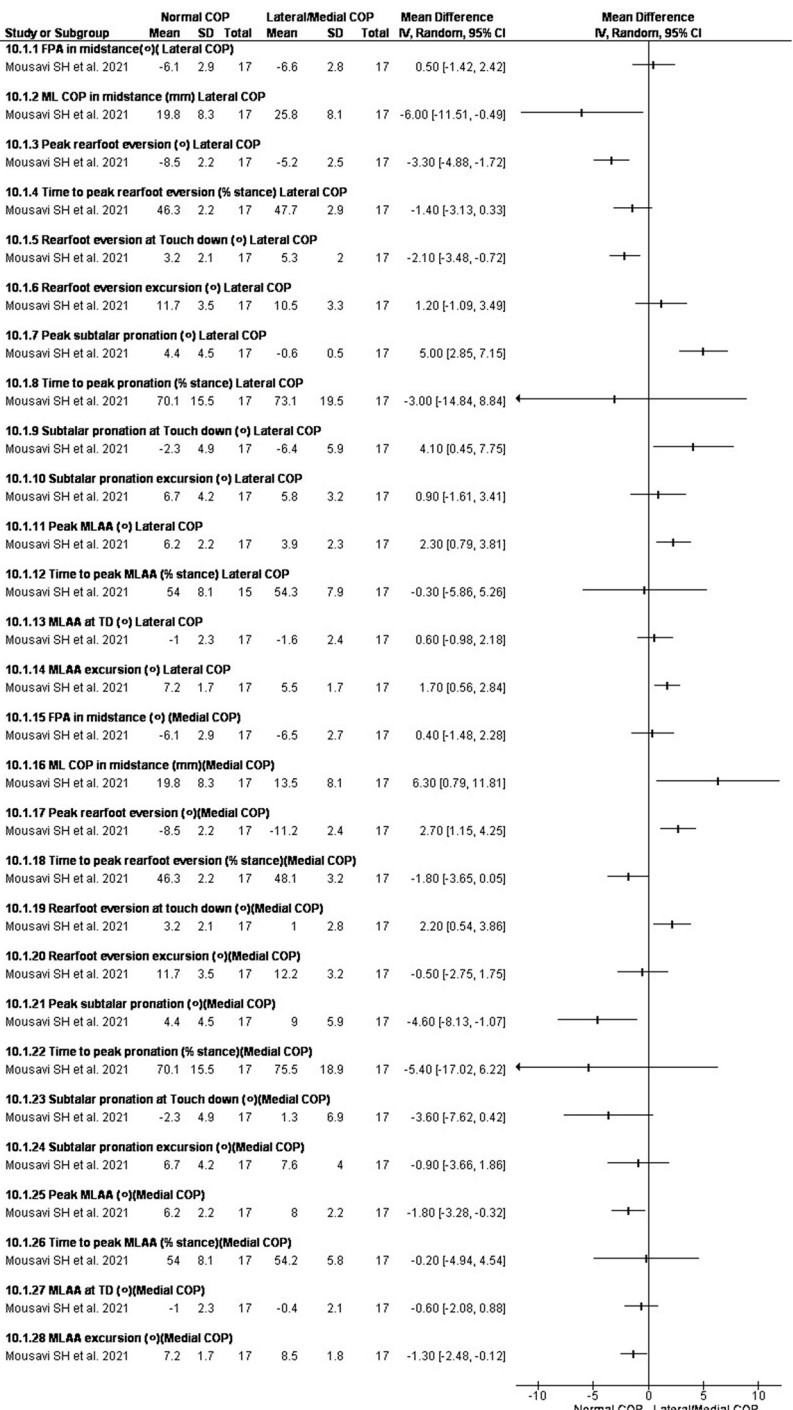

**Fig 9. Results of changing COP on rearfoot eversion and MLAA.**

Backward walking (BW) significantly decreased right and left Foot posture index (FPI) by -0.50 [-0.91, -0.09] and -0.95 [-1.44, -0.46] scores compared to control group [53] (Fig 11).

Fig 12 shows the graphical abstract of the effects of gait retraining modifications on foot pronation.

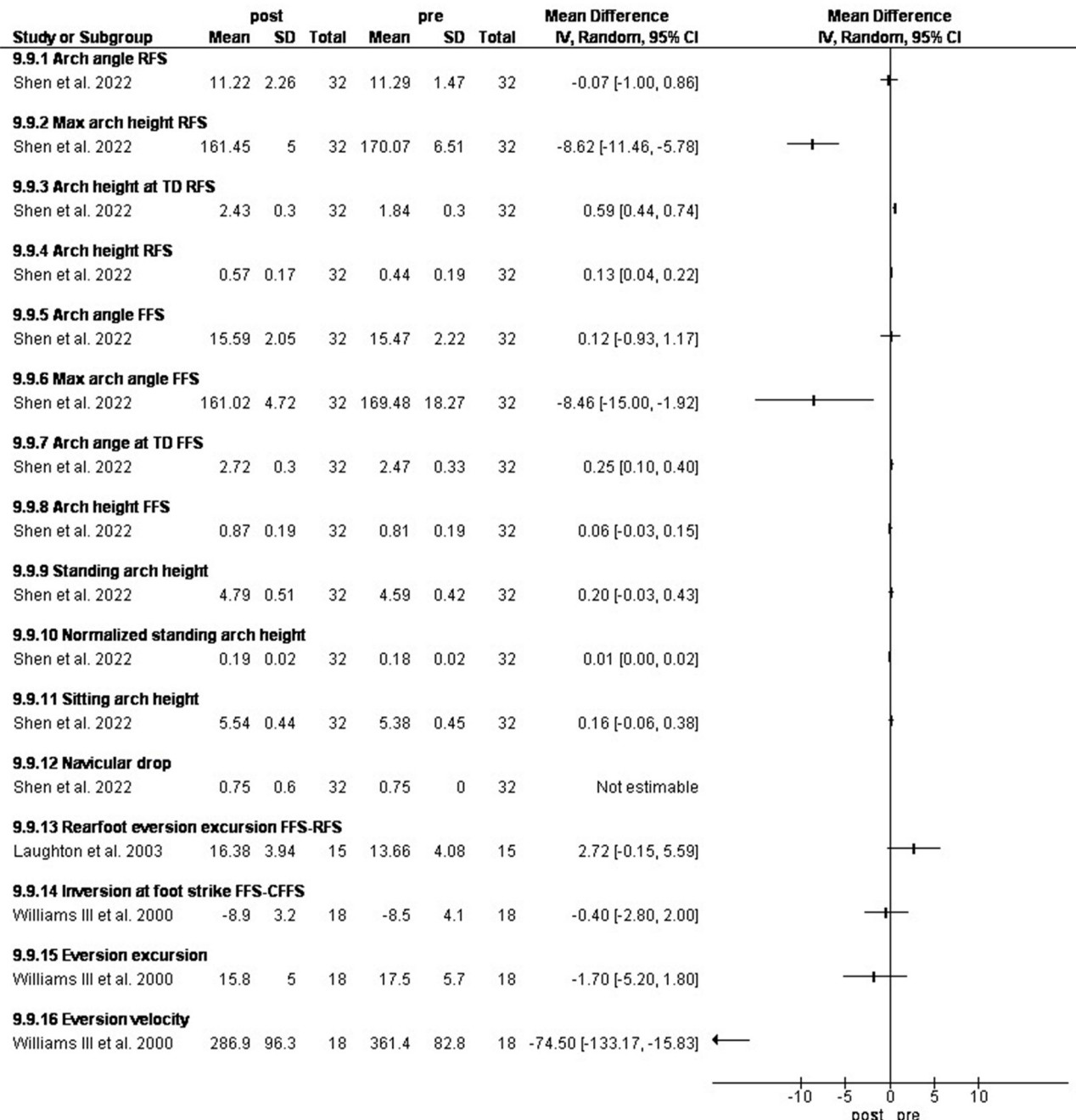

**Fig 10. Results of changing foot strike on rearfoot eversion and MLAA.**

## Discussion

### Changing foot progression angle

Two studies assessed changing foot progression angle. In study 1 [17], toe-in was able to alleviate foot over-pronation without exhibiting any discomfort. In study 2 [55], toe-in running using real-time visual feedback reduced peak rearfoot eversion, peak pronation, and peak

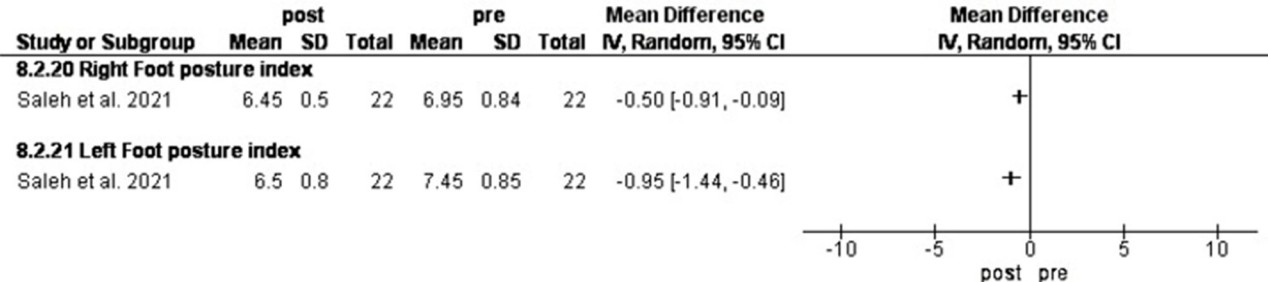

**Fig 11. Results of backward walking on foot posture.**

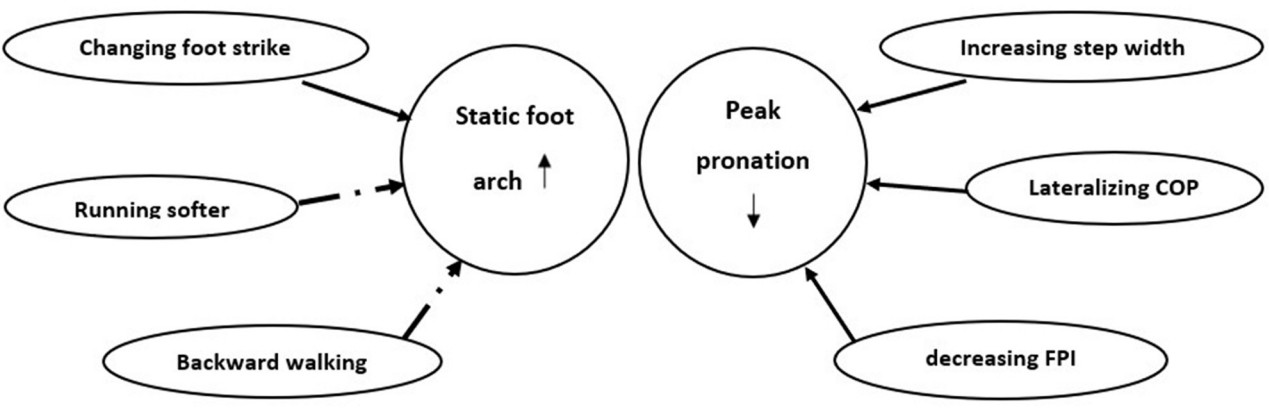

(L); large effect size, (M); moderate effect size, (s); small effect size.

Moderate evidence;——, limited evidence; —·►

**Fig 12. Graphical abstract of the effects of gait retraining on dynamic peak pronation and static foot arch.**

MLAA and increased internal longitudinal arch compared to normal and toe-out running. These 2 studies showed that the foot is more supinated when toeing-in [55,62]. Thus, in individuals with over-pronation, toeing-in not only may reduce foot pronation, it may contribute to foot stabilization at touchdown and late stance phase of walking.

MLAA is also associated with FPA when running and walking. Study 2 reported about a 1-degree change in the arch of the foot after a change of 5 degrees in the angle of the foot, which can be doubted for the required amount of change from a clinical point of view [55]. Although, by changing the angle of the foot, how much the internal longitudinal arch can change in people with flat feet, needs to be investigated.

## Changing speed

Three studies assessed the effect of changing speed [54,61,64] on foot pronation. In the study by Pohl et al. [54], rearfoot eversion excursion and time to peak eversion was only significantly different between walking and running. In the study by Farina and Hahn [64], increasing step

rate by 5 to %10 significantly decreased peak rearfoot eversion. A previous study [65] assessed the effect of increasing step rate in which the decrease in rearfoot eversion was not significant. These studies used an overground running protocol in which the preferred running step rate was lower than in the included study in our systematic review.

In the study by Dunn et al. [61], faster absolute speed (ABS) and relative (REL) speed for running retraining increased foot strike index but did not change peak rearfoot eversion. Increasing step rate has been shown to decrease peak rearfoot eversion [66]. With +5% and +10% of preferred step rate, frontal rearfoot eversion was decreased.

## Running softer

With running softer, AI was increased. After static training intervention, the FPI showed a decrease in the foot pressure and led to better support of the dynamic plantar AI in recreational runners [52].

Studies are required to compare the efficacy of different types of feedback and various schedules. A recent study [67] showed that auditory biofeedback and external focus of attention compared to internal focus can lead to more improvements alleviating the biomechanical factors associated with ankle instability. This included study [52] evaluated gait retraining with visual biofeedback in no-injury recreational runners, however, the results are promising for diminishing the risk factors involved in RRIs for example by reducing plantar load and foot pronation (using visual biofeedback for running softer).

Moreover, plantar load under the heel was decreased utilizing a plantar pressure distribution in a natural environment rather than running on a treadmill [68,69]. Thus, the results can be implemented in clinical setting.

## Changing mediolateral COP

Lateralizing the COP reduced pronation in both included studies. In study 1 [17], running with more lateral COP reduced peak rearfoot eversion and excursion, peak subtalar pronation, peak MLAA compared to normal running. On the other hand, medializing the COP increased the abovementioned variables. In study 2 [51], peak ankle eversion was decreased by lateralizing the COP peak. This study reported that lateralizing the COP can increase peak ankle inversion while no significant change was found in peak ankle moment for medial shift.

Based on the results of these 2 studies it can be derived that the control of medializing the COP is more difficult to perform which can be due to tighter tissues of the medial-foot. As lateralizing the COP decreased foot pronation, it may affect tibialis posterior which is the main muscle for contorting subtalar pronation. Therefore, lateralizing COP can be a way to better activation of the posterior tibialis in those suffering from excessive foot pronation with the origin of posterior tibialis deficit.

The study [56] showed that lateralizing or medializing foot pressure without feedback while running, affects the FPA. However, participants were adopted to the experiment by familiarization period with real-time visual feedback in which the pointer was aligned with the subject's FPA.

However, it should be considered that lateralizing COP may lead to increased risk of injuries (e.g., stress fracture in the foot [70] and lower leg overuse injuries [71]). Therefore, clinicians and researchers are advised to consider the side effects of manipulating mediolateral COP for rearfoot eversion modification, while designing a prevention or treatment program.

## Changing step width

Increasing step width decreased rearfoot eversion [41,42] and excursion angles and decreased rearfoot inversion moment [41] and foot pronation and maximal pronation velocity [50].

By increasing step width, peak rearfoot eversion angle decreased about 1–2 degrees [41,42] with a greater change in men than women [41]. The study by William and Ziff [50] showed that a 1° change in peak pronation angle was caused by just a 1–3 cm change in step width.

The change in rearfoot eversion following increasing step width is thought to be due to the fact that wider steps create greater stability and decrease the amount of internal rotation of the leg. A change in step width is something that can be easily tried with minimal time and effort and it appears that a relatively minor change can bring about a change in rearfoot pronation similar to what might be achieved by changing footwear or using orthotics.

It should be realized that changing step width could have both promising and detrimental effects, particularly if the changes are extreme. While altering step width can have a positive effect on modifying foot pronation, resulting changes in internal stresses could exacerbate some symptoms instead of relieving them. Therefore, the ideal step width for reducing foot pronation and rearfoot eversion is still up for debate. If an individual aims to relieve over-pronation symptoms by adjusting their step width, it is crucial to implement gradual, minor adjustments over time to allow the body to adapt.

## Changing foot strike

In study 1, FFS training significantly increased the standing arch height. eversion excursion was also significantly increased due to greater amounts of inversion at foot strike with the FFS pattern, with moderate effect size (Cohen's d = 0.55) [45].

According to Shen et al., [43], a 12-week gait retraining program, along with foot core exercise, resulted in improvements in foot pronation and navicular height in both standing and dynamic positions. These improvements were found to have a moderate to large effect size [43]. During rear foot strike (RFS) running, such intervention decreased the maximum arch angle and increased arch height at touchdown, During RFS, the peak arch angle reduced by 5.1% while the arch height increased at touchdown by 32.1%.

In the study by Williams et al. [44], rearfoot kinematics were not significantly different between forefoot strikers and rearfoot strikers who were instructed for forefoot strike. The CFFS were plantarflexed and inverted at foot strike. 'They also demonstrated similar rearfoot eversion excursions. Finally, eversion velocities of the CFFS were similar to the FFS. The rearfoot becomes supinated as it plantarflexes and therefore will be associated with the greater inversion at foot strike. As with the FFS, the CFFS reached similar peak eversion values resulting in larger excursions in these planes of motion, with associated increases in eversion velocities, thus potentially increasing the demands of the muscles necessary to control these motions.

The arches in the foot, both longitudinal and transverse, contain elastic tissues that can restore approximately 17% of the mechanical energy produced while walking [72]. Therefore, it can be suggested that FFS may serve as a valuable training technique to improve arch performance [73]. However, some researchers have suggested that this intervention may not be suitable for all individuals due to the potential risk of other injuries, such as Achilles tendonitis and calf strains, as well as the need for significant changes in running form. A previous study reported that rearfoot strikers are instructed to run with an FFS pattern, they do not differ in rearfoot kinematics from natural forefoot strikers [44]. Besides, FFS runners have been shown to have increased rearfoot eversion excursions and velocities compared to RFS runners [74].

## Backward walking training

Backward walking training (BW) reduced the FPI. BW as an addition to physical therapy exercises of flat foot can improve foot posture in comparison with physical therapy exercises alone and requires greater activation of the responsible muscles for foot and ankle control, which can enhance proprioception and improve overall foot mechanics. Additionally, BW can promote a more upright posture when assessed by the FPI in long-distance runners, which can contribute to optimal foot position. Also, the applied exercises could have beneficial effects on foot alignment [75].

BW training's superiority is its justifiability as it involves an unusual pattern with toes contacting first and the heel finishing the step, leading to greater attention and cautiousness due to instability. As a result, motor cortex activity increases by 30% [76], promoting postural stability [53].

Altogether, almost all gait-retraining methods had a significant impact on foot pronation. Previously, orthotics demonstrated 1–2 degrees of change in pronation angle which can alleviate adverse symptoms. Though, finding the right shoes or orthotics can be costly and time-consuming [50]. Therefore, gait retraining can be an advantageous surrogate. However, it is difficult to determine which method of gait retraining is the "best" as results may vary depending on the individual and the specific condition being treated. Therefore, it is recommended that gait-retraining should be tailored to the individual's specific needs and goals, with the guidance of a qualified healthcare professional to ensure optimal outcomes.

## Limitations and recommendations for future studies

This study identified several limitations. Firstly, the long-lasting effects of running retraining could not be investigated due to the maximum intervention time of 6 hours. Future studies assessing long-term effects are required. Secondly, some interventions were assessed only on male or female subjects with uncontrolled strike patterns, and only one study had participants with pronated foot. Further research on male and female runners and non-runners with pronated feet and different foot strike patterns in separated groups with controlled previous and other sports injuries is warranted. Thirdly, due to the small sample sizes in some studies, further studies including high-quality RCTs with appropriate methodology are needed, such as calculated sample sizes, random sampling, concealment of subjects and assessors, and adjustment of confounding factors in statistical analysis. Furthermore, future studies assessing the biomechanics of lower limbs at various running speeds are needed, with higher frequencies of data collection for higher speeds.

Assessing the effects of gait retraining on foot pronation, considering the propulsion and pros and cons of manipulating gait with different methods, is recommended. In addition, gait should be investigated in both laboratory and real-world settings to assess daily life, and controlling external factors affecting gait. For real-world assessment, 2D measurement, smartphone applications, and shoe-embedded sensors for walking can be utilized. Standardized shoes and proper marker attachment should be taken into account as factors that influence the results. Moreover, different outcomes predicting foot pronation, such as electromyography, should be measured, and biofeedback should be given to change mediolateral COP, taking into consideration the cognitive overload due to the complexity of adjusting movement for two feet each in two degrees of freedom. Utilizing either larger target quadrants of the foot or visual feedback of the displacement only along the anteroposterior or mediolateral axis should be considered, and the effects of different muscle activity on foot pronation during gait should be investigated. Foot core exercise can be added to all variations of gait retraining programs for those with weak arch muscles.

## Conclusion

In conclusion, the findings of this systematic review suggest that gait retraining can be an effective intervention to reduce foot pronation. Most of the included studies demonstrated significant improvements in foot pronation following gait retraining, including changing COP, step width, foot progression angle, speed, foot strike, running softer and backward walking. Changing center of pressure, foot progression angle and forefoot strike training appeared to yield more favorable outcomes. However, more evidence is required to conclude the effects of each gait retraining on foot pronation.

Previously, orthotics showed significant results but they make dependency and are costly. In contrast, gait retraining is easily implemented and low-cost and superior to foot orthoses. However, the optimal type and dosage of gait retraining remain unclear. Further research is warranted to investigate the long-term effects of gait retraining on foot pronation. Clinicians are advanced to use gait retraining in the management of abnormal foot pronation.

## Supporting information

**S1 Table. Prisma checklist.**
(DOCX)

## Author Contributions

**Conceptualization:** Seyed Hamed Mousavi, Fateme Khorramroo, Amirali Jafarnezhadgero.

**Data curation:** Seyed Hamed Mousavi, Fateme Khorramroo.

**Formal analysis:** Seyed Hamed Mousavi, Fateme Khorramroo, Amirali Jafarnezhadgero.

**Investigation:** Seyed Hamed Mousavi, Fateme Khorramroo.

**Methodology:** Seyed Hamed Mousavi, Fateme Khorramroo, Amirali Jafarnezhadgero.

**Project administration:** Seyed Hamed Mousavi, Fateme Khorramroo, Amirali Jafarnezhadgero.

**Supervision:** Seyed Hamed Mousavi, Amirali Jafarnezhadgero.

**Validation:** Seyed Hamed Mousavi, Fateme Khorramroo, Amirali Jafarnezhadgero.

**Visualization:** Fateme Khorramroo.

**Writing – original draft:** Fateme Khorramroo.

**Writing – review & editing:** Seyed Hamed Mousavi, Amirali Jafarnezhadgero.

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
