## [Decision Letter · Decision Letter 0]

26 Dec 2023

PONE-D-23-38212Gait Retraining Targeting Foot Pronation: A Systematic Review and Meta-analysis

PLOS ONE

Dear Dr. Khorramroo,

Thank you for submitting your manuscript to PLOS ONE. After careful consideration, we feel that it has merit but does not fully meet PLOS ONE’s publication criteria as it currently stands. Therefore, we invite you to submit a revised version of the manuscript that addresses the points raised during the review process.

Dear Author;

I trust this letter finds you well. Thank you for submitting your manuscript titled "[Gait Retraining Targeting Foot Pronation: A Systematic Review and Meta-analysis]" to [PLOS ONE]. I appreciate the effort you and your co-authors have invested in this work. After a careful review, the decision has been made that the manuscript requires major revisions before it can be considered for publication in [PLOS ONE].

The research is intriguing and well-organized; however, specific comments and limitations have been identified that need attention to enhance the overall quality of the submission. In the comments section, it was noted that the tables in the results are not clear, warranting a thorough review and improvement to ensure they are well-organized, easily interpretable, and properly formatted. Additionally, comprehensive descriptions of all tests, including backward walking training, changing foot strike, etc., are crucial before presenting the results to provide readers with a clear understanding of the procedures and context of the conducted tests.

Furthermore, the absence of images in the review was observed. Incorporating relevant images could significantly enhance the overall clarity of the research. Please consider adding visual elements to illustrate key points and support the textual content effectively.

In the limitations section, it was reiterated that addressing the clarity issues associated with tables, providing comprehensive descriptions of all tests, and including relevant images are crucial steps toward refining the manuscript. Your attention to these aspects will undoubtedly contribute to the overall improvement of the research.

I trust that you will consider these comments carefully during the revision process and incorporate responses to these concerns in your cover letter upon resubmission. If you have any questions or require further clarification, please do not hesitate to reach out. Thank you for your understanding and cooperation. I look forward to reviewing the revised version of your manuscript.

Please submit your manuscript on Feb 09 2024 11:59PM.

Sincerely,

Changes in Distance Running Mechanics Due to Systematic Variations in Running Style - https://doi.org/10.1123/ijsb.7.1.76

Effectiveness of Lower-Cost Strategies for Running Gait Retraining: A Systematic Review - https://doi.org/10.3390/app13031376

(among others)

In your revision ensure you cite all your sources (including your own works), and quote or rephrase any duplicated text outside the methods section. Further consideration is dependent on these concerns being addressed.

3. Please include your tables as part of your main manuscript and remove the individual files. Please note that supplementary tables (should remain/ be uploaded) as separate ""supporting information"" files

Reviewers' comments:

Reviewer's Responses to Questions

**Comments to the Author**

1. Is the manuscript technically sound, and do the data support the conclusions?

Reviewer #1: No

Reviewer #2: Yes

2. Has the statistical analysis been performed appropriately and rigorously? 

Reviewer #1: No

Reviewer #2: I Don't Know

3. Have the authors made all data underlying the findings in their manuscript fully available?

Reviewer #1: No

Reviewer #2: No

4. Is the manuscript presented in an intelligible fashion and written in standard English?

Reviewer #1: No

Reviewer #2: Yes

5. Review Comments to the Author

Reviewer #1: This manuscript requires a significant amount of improvement in

Clarify the known and unknown of your review

What is the novelty of your review?

All figures are not clear, they needed readjustment and enhancement

Tables not found even in the manuscript or the supplementary documents

Why you did not have Prospero registration for the study

Clarify you are using the Downs and black scale for Quality assessment

Abstract:

Include a brief and concise background of the study.

The database searched was very limited. (only 4 )

Include the character of the study articles.

Mention the statistical tests used for the analysis and reports with SMD and CI for the outcome

The conclusion should be more concise and clear based on the study reports.

Introduction

very long introduction, you should make it concise and brief and highlight the previous reviews and the unique aims of the current review depending on the limitations and recommendations of the previous studies

Inclusion and exclusion criteria not satisfied,

you should formulate inclusion criteria depending on the types of PICO, patients ,intervention, comparison , outcomes

results and discussion need to be rewritten using Prisma guidelines to be more clear

References are very old, it was more than 20 years (as an example references no 1,2,3,4,5,6,12,14,48,52,60,63,69,76,85)

Regarding Reference's full style, You need to check the author's instructions

Reviewer #2: The research is interesting and organized, but there are some comments.

Limitations:

1. The tables in the results are not clear.

2. All of the tests mentioned need to be described before the results are presented (backward walking training, changing foot strike, etc.).

3. The review misses the presence of images

6. PLOS authors have the option to publish the peer review history of their article (what does this mean?). If published, this will include your full peer review and any attached files.

Reviewer #1: No

Reviewer #2: No

While revising your submission, please upload your figure files to the Preflight Analysis and Conversion Engine (PACE) digital diagnostic tool, https://pacev2.apexcovantage.com/. PACE helps ensure that figures meet PLOS requirements. To use PACE, you must first register as a user. Registration is free. Then, login and navigate to the UPLOAD tab, where you will find detailed instructions on how to use the tool. If you encounter any issues or have any questions when using PACE, please email PLOS at figures@plos.org. Please note that supporting Information files do not need this step.

---

## [Author Response · Author response to Decision Letter 0]

22 Jan 2024

Dear editor and reviewers;

We thank you for your insightful and constructive suggestions. Based on your helpful suggestions, we were able to further improve our manuscript. We carefully considered and addressed all your specific comments and revised the text if necessary. 

Please find your comments in bold and our responses in italic font. In the manuscript, all the asked changes are highlighted as it appears in this file (Response to reviewers) or track changed.

Comments from the editors and reviewers:

-Editor

I trust this letter finds you well. Thank you for submitting your manuscript titled "[Gait Retraining Targeting Foot Pronation: A Systematic Review and Meta-analysis]" to [PLOS ONE]. I appreciate the effort you and your co-authors have invested in this work. After a careful review, the decision has been made that the manuscript requires major revisions before it can be considered for publication in [PLOS ONE].

The research is intriguing and well-organized; however, specific comments and limitations have been identified that need attention to enhance the overall quality of the submission. In the comments section, it was noted that the tables in the results are not clear, warranting a thorough review and improvement to ensure they are well-organized, easily interpretable, and properly formatted. Additionally, comprehensive descriptions of all tests, including backward walking training, changing foot strike, etc., are crucial before presenting the results to provide readers with a clear understanding of the procedures and context of the conducted tests.

Furthermore, the absence of images in the review was observed. Incorporating relevant images could significantly enhance the overall clarity of the research. Please consider adding visual elements to illustrate key points and support the textual content effectively.

In the limitations section, it was reiterated that addressing the clarity issues associated with tables, providing comprehensive descriptions of all tests, and including relevant images are crucial steps toward refining the manuscript. Your attention to these aspects will undoubtedly contribute to the overall improvement of the research.

Please submit your manuscript on Feb 09 2024 11:59PM.

We would like to thank you and your team for the efficient handling of the review process. We are pleased to inform you that we have thoroughly addressed all the comments and questions raised by the reviewers, ensuring that our manuscript is now more robust and comprehensive. We increased the quality of figures and tables, added the description of all interventions in each section of the Results and added a graphical abstract for the effect of modifications on foot pronation and static foot arch.

Reviewers’ comments

-Reviewer #1

This manuscript requires a significant amount of improvement in

Clarify the known and unknown of your review

Thank you for your comment. We explained in the conclusion as bellow:

In conclusion, the findings of this systematic review suggest that gait retraining can be an effective intervention to reduce foot pronation. Most of the included studies demonstrated significant improvements in foot pronation following gait retraining, including changing COP, step width, foot progression angle, speed, foot strike, running softer and backward walking. Changing center of pressure, foot progression angle and forefoot strike training appeared to yield more favorable outcomes. However, more evidence is required to conclude the effects of each gait retraining on foot pronation.

Previously, orthotics showed significant results but they make dependency and are costly. In contrast, gait retraining is easily implemented and low-cost and superior to foot orthoses. However, the optimal type and dosage of gait retraining remain unclear. Further research is warranted to investigate the long-term effects of gait retraining on foot pronation. Clinicians are advanced to use gait retraining in the management of abnormal foot pronation.

What is the novelty of your review?

Thank you for your comment. Our research is unique in that we conducted a thorough search of 4 databases and google scholar to investigate the effect of gait retraining on foot pronation. After obtaining 15 studies, we found that no systematic reviews had been conducted on this topic. Therefore, we decided to provide a systematic review to use its results, recommendations and find the gaps to conduct future studies. We added the following sentences:

Although several studies have assessed the effects of gait retraining on foot pronation, no systematic review synthesizing the evidence on this topic has been published. Therefore, this systematic review and meta-analysis aimed to explore the effect of gait retraining targeting foot pronation. Potential limitations and future research directions are discussed to guide clinical practice and future investigation.

All figures are not clear, they needed readjustment and enhancement

Thank you for your comment. We improved the quality of all figures.

Tables not found even in the manuscript or the supplementary documents

Thank you for your comment. As requested by the journal, all table were submitted in a separate file. Probably the journal had not sent them to the reviewers. So, we inserted all the tables in the revised manuscript file.

Why you did not have Prospero registration for the study

All our registrations were being rejected automatically with this message: “To enable PROSPERO to focus on COVID-19 registrations during the pandemic, this registration record was automatically rejected because it did not meet the acceptance criteria.” However, we found out that PROSPERO will solve this issue, if we email them; however, our paper was almost finished at that time.

Clarify you are using the Downs and black scale for Quality assessment

Thank you for your comment. We added in the text that why we have used downs and black and inserted the quality assessment table in the manuscript. We added the following text in the manuscript:

Two authors (FKH and SHM ) independently assessed the methodological quality of the included studies using the modified Downs and Black checklist [48]. The complete form was used to assess RCTs and 15 questions were used to assess non-RCTs. Any disparities in scoring were rechecked and if necessary, a consensus was reached using the third reviewer (AJG). 

Abstract:

Include a brief and concise background of the study.

Thank you for your comment. We added more explanation in the background part of the abstract as bellow. 

Foot pronation is a prevalent condition known to contribute to a range of lower extremity injuries. Numerous interventions have been employed to address this issue, many of which are expensive and necessitate specific facilities. Gait retraining has emerged as a promising intervention for mitigating foot pronation, offering the advantage of being accessible and independent of additional materials or specific time.

The database searched was very limited. (Only 4)

Thank you for your consideration and accountability. We searched google scholar to ensure if there are any missed studies from retrieved ones from those 4 databases and we found no additional study. The following published systematic reviews have also searched the same databases.

Kinematic risk factors for lower limb tendinopathy in distance runners: A systematic review and meta-analysis

The relationship between static and dynamic foot posture and running biomechanics: A systematic review and meta-analysis

Include the character of the study articles.

Thank you for your suggestion. We added a description of the intervention in the results section additional to the information provided in quality assessment and data extraction tables which are now inserted in the manuscript.

Mention the statistical tests used for the analysis and reports with SMD and CI for the outcome

Thank you for your careful review. We added statistical analysis in the Methods section as bellow:

Mean differences and 95% confidence intervals (CI) were calculated with random effects model in RevMan version 5.4. 

The conclusion should be more concise and clearer based on the study reports.

Thank you for your suggestion. We added more details as bellow:

Overall, this study suggests that gait retraining may be a promising intervention for reducing foot pronation; Most of the included studies demonstrated significant improvements in foot pronation following gait retraining. Changing center of pressure, foot progression angle and forefoot strike training appeared to yield more favorable outcomes. However, further research is needed to fully understand its effectiveness and long-term benefits.

Introduction

very long introduction, you should make it concise and brief and highlight the previous reviews and the unique aims of the current review depending on the limitations and recommendations of the previous studies

Thank you for your comment. We summarized the Introduction and made it more concise and briefer. We mentioned previous studies as bellow:

Gait retraining is increasingly utilized as a novel way of inducing the body or a segment to alter a movement pattern or a segment’s motion direction [31]. There is a variety of techniques from easy (e.g., manipulating step rate) to complex (e.g., tibial acceleration decrease) for gait retraining [32]. Recent studies have suggested gait training to change the lower limb biomechanics [28–31,33,34]. The most frequently modified parameters for retraining include step rate, step width and foot strike pattern [34–36].

Several studies have proposed changes to running techniques (i.e., movement) through running retraining using feedback to reduce impact loads [37]. A study found that increasing step cadence by just 5% significantly reduced peak braking force by 5.7% [38] and 11.4% [39] in long-distance runners. Increasing step cadence with a proportional reduction in the stride length at a constant speed has reduced foot inclination angles and impact forces by 5.6% [40] which decreases the number of initial contacts by hindfoot[41]. Besides, altering step width has reduced foot pronation [42,43]. Forefoot strike training has also demonstrated promising results in increasing foot arch [44–46].

Although several studies have assessed the effects of gait retraining on foot pronation, no systematic review synthesizing the evidence on this topic has been published. Therefore, this systematic review and meta-analysis aimed to explore the effect of gait retraining targeting foot pronation. Potential limitations and future research directions are discussed to guide clinical practice and future investigation.

Inclusion and exclusion criteria not satisfied,

you should formulate inclusion criteria depending on the types of PICO, patients, intervention, comparison, outcomes

Thank you for your careful review. We completed the inclusion and exclusion criteria as bellow:

The inclusion criteria were: Written-English studies comparing the effect of gait retraining before and after interventions on foot pronation, in studies that included participants with either supinated or pronated foot or participants without any abnormality in the foot arch. 

The exclusion criteria were: non-English studies, studies with an intervention other than gait retraining or assessing effects other than foot pronation or investigated on individuals with specific abnormalities such as knee valgus.

results and discussion need to be rewritten using Prisma guidelines to be more clear

Thank you for your comment. We edited the results and discussion trying to mention all the required information in the PRISMA. The changes were highlighted in the manuscript.

References are very old, it was more than 20 years (as an example references no 1,2,3,4,5,6,12,14,48,52,60,63,69,76,85)

Regarding Reference's full style, you need to check the author's instructions

Thank you for your helpful suggestions. We replaced some new references. 

-Reviewer #2

The research is interesting and organized, but there are some comments.

Thank you for your review and positive feedback. We are grateful for the opportunity to improve our paper under your guidance.

Limitations:

1. The tables in the results are not clear.

Thank you for your comment. We increased the quality of all the figures and tables and included them in the manuscript.

2. All of the tests mentioned need to be described before the results are presented (backward walking training, changing foot strike, etc.).

Thank you for your careful suggestion. We added the description of the interventions included in our systematic review in the Results section before reporting the results which are highlighted.

3. The review misses the presence of images

Thank you for your comment. We added all the figures and tables in the manuscript. Moreover, we inserted a graphical abstract to show the overall effect of gait retraining on foot pronation and static foot arch.

---

## [Editor Report · Decision Letter 1]

29 Jan 2024

Gait Retraining Targeting Foot Pronation: A Systematic Review and Meta-analysis

PONE-D-23-38212R1

Dear Dr. Khorramroo,

We’re pleased to inform you that your manuscript has been judged scientifically suitable for publication and will be formally accepted for publication once it meets all outstanding technical requirements.

Kind regards,

Ateya Megahed Ibrahim El-eglany

Academic Editor

PLOS ONE
---

## [Editor Report · Acceptance letter]

21 Feb 2024

PONE-D-23-38212R1 

PLOS ONE

Dear Dr. Khorramroo, 

I'm pleased to inform you that your manuscript has been deemed suitable for publication in PLOS ONE. Congratulations! Your manuscript is now being handed over to our production team.

Kind regards, 

on behalf of

Dr. Ateya Megahed Ibrahim El-eglany 

Academic Editor

PLOS ONE